# Do Surgeons Anticipate Women’s Hopes and Fears Associated with Prolapse Repair? A Qualitative Analysis in the PROSPERE Trial

**DOI:** 10.3390/jcm12041332

**Published:** 2023-02-07

**Authors:** Xavier Fritel, Marion Ravit, Anne-Cécile Pizzoferrato, Sandrine Campagne-Loiseau, Georges Bader, Perrine Capmas, Michel Cosson, Philippe Debodinance, Xavier Deffieux, Hervé Fernandez, Philippe Ferry, Olivier Garbin, Bernard Jacquetin, Guillaume Legendre, Christian Saussine, Renaud de Tayrac, Laurent Wagner, Jean-Philippe Lucot, Arnaud Fauconnier

**Affiliations:** 1Faculté de Médecine et Pharmacie, Université de Poitiers, INSERM CIC 1402, Service de Gynécologie-Obstétrique et Médecine de la Reproduction, CHU de Poitiers, 86000 Poitiers, France; 2Emergency Obstetric and Quality of Care Unit, Liverpool School of Tropical Medicine, Liverpool L3 5QA, UK; 3Service de Gynécologie, CHU Estaing, 63000 Clermont-Ferrand, France; 4Medical and Surgical Center Ambroise Paré, Hartmann, Pierre Cherest, 92200 Neuilly-sur-Seine, France; 5INSERM Centre of research in Epidemiology and population health (CESP), U1018, 94276 Villejuif, France; 6Service de Gynécologie-Obstétrique, Hôpital Bicêtre, GHU Sud, AP-HP, 94270 Le Kremlin-Bicêtre, France; 7Faculté de Médecine, Université de Lille, Service de Gynécologie, CHRU Jeanne de Flandre, 59000 Lille, France; 8Dunkerque Hospital Center, 59240 Dunkerque, France; 9Faculté de Médecine, Université Paris-Sud, Service de Gynécologie-Obstétrique, Hôpital Antoine-Béclère APHP, 92140 Clamart, France; 10La Rochelle Hospital Center, 17000 La Rochelle, France; 11Faculté de Médecine, Université de Strasbourg, Service de Gynécologie, CHRU de Strasbourg, 67000 Strasbourg, France; 12Faculté de Médecine, Université d’Angers, UMR S1085, Service de Gynécologie-Obstétrique, CHU d’Angers, 49000 Angers, France; 13Faculté de Médecine, Université de Strasbourg, Service d’Urologie, CHRU de Strasbourg, 67000 Strasbourg, France; 14Faculté de Médecine, Université de Montpellier, Service de Gynécologie-Obstétrique, CHU Carémeau, 30000 Nîmes, France; 15Service d’Urologie, CHU Carémeau, 30000 Nîmes, France; 16Saint-Vincent-de-Paul Hospital Center, 59000 Lille, France; 17Faculté de Médecine, Université Paris-Saclay, Service de Gynécologie-Obstétrique, CHI Poissy-Saint-Germain-en-Laye, 78000 Poissy, France

**Keywords:** pelvic organ prolapse, surgery, expectations, hope, fear

## Abstract

Women’s preoperative perceptions of pelvic-floor disorders may differ from those of their physicians. Our objective was to specify women’s hopes and fears before cystocele repair, and to compare them to those that surgeons anticipate. We performed a secondary qualitative analysis of data from the PROSPERE trial. Among the 265 women included, 98% reported at least one hope and 86% one fear before surgery. Sixteen surgeons also completed the free expectations-questionnaire as a typical patient would. Women’s hopes covered seven themes, and women’s fears eleven. Women’s hopes were concerning prolapse repair (60%), improvement of urinary function (39%), capacity for physical activities (28%), sexual function (27%), well-being (25%), and end of pain or heaviness (19%). Women’s fears were concerning prolapse relapse (38%), perioperative concerns (28%), urinary disorders (26%), pain (19%), sexual problems (10%), and physical impairment (6%). Surgeons anticipated typical hopes and fears which were very similar to those the majority of women reported. However, only 60% of the women reported prolapse repair as an expectation. Women’s expectations appear reasonable and consistent with the scientific literature on the improvement and the risk of relapse or complication related to cystocele repair. Our analysis encourages surgeons to consider individual woman’s expectations before pelvic-floor repair.

## 1. Introduction

Pelvic organ prolapse (POP) is a frequent but not life-threatening disorder. It is an important cause of morbidity, and has a well-documented negative impact on quality of life (QoL) [1]. Some women affected by this condition may choose surgical repair, while others may not. A woman’s lifetime risk of prolapse surgery is estimated at 19% [2].

Little is known about why women choose surgical treatment and their expectations of surgical repair. The severity of symptoms probably contributes to seeking this surgery [3]. Data on women’s fears before prolapse surgery are even sparser. Complications reported with the use of mesh may induce specific fears.

Women’s preoperative perceptions of the problems associated with this disorder appear to differ from those of their physicians [4]. It may be that women have greater expectations—higher hopes—for the results of surgical repair. We do not know the extent to which surgeons may anticipate women’s expectations.

Surgeons habitually assess the results of their interventions based on the extent of the anatomical correction [5], the improvement of symptoms such as protrusion, and functional urinary or rectal signs [6]. More and more authors also analyze the results of prolapse surgery with a QoL scale. Nonetheless, these scales do not include women’s expectations of the surgery—expectations themselves influenced by their past experiences. Understanding and identifying these expectations better before the intervention could enable an increase in women’s satisfaction.

Today, when surgery with mesh implants is increasingly called into question and access to health information massively facilitated by the diversity and variability of its sources, it is necessary to tailor the management, information, and advice provided as part of shared medical decision making, which is a major advance in the management of functional disorders [7].

Our aim was to perform a secondary qualitative analysis of expectations (hopes and fears) reported by women before surgery scheduled for pelvic organ prolapse (POP) and included in the PROSPERE (PROSthetic PElvic floor REpair) [8] randomized-controlled trial, and to compare the hopes and fears reported by the women to those that surgeons anticipate their patients will have.

## 2. Materials and Methods

The PROSPERE multicenter randomized trial was designed to compare complications associated with cystocele repair with mesh by laparoscopy versus transvaginally [8]. The study of women’s hopes and fears before surgery was planned as an ancillary study before the trial began.

### 2.1. Inclusion

From October 2012 through April 2014, participating surgeons invited patients aged 45 to 75 years who had a symptomatic prolapse of the anterior vaginal wall stage 2 or higher (according to the Pelvic Organ Prolapse Quantification (POP-Q) system) to participate in this study. Exclusion criteria were previous surgical POP repair, the impossibility of, or a contraindication to, either surgical route, pelvic-organ cancer, contraindication to the use of mesh, inability to read French, lack of health insurance, pregnancy, or a desire to become pregnant in the future.

The study-information leaflet given to each woman before requesting her written consent explained that complications were the principal outcome measure of the PROSPERE study—both intraoperative and postoperative, during the year after surgery. The intervention principles were described. The document listed as possible complications of the surgery its potential failure, vaginal, gastrointestinal, or bladder exposure of the mesh implant, bladder injury, infection, intestinal injury, hemorrhage, chronic pain, dyspareunia, constipation, occlusion, incontinence, or urinary symptoms.

### 2.2. Questionnaires

Before surgery, we asked each woman included to report 5 hopes and 5 fears (ranked in descending order of importance) that she had concerning the planned operation [9]. Women who mentioned at least one hope or one fear on the inclusion questionnaire were included in our analysis. We also asked the surgeons (blinded to the women’s responses) involved in the RCT to report and rank what they presumed would be the preoperative hopes and fears of a typical patient.

### 2.3. Data Analysis

Categorization of women’s hopes and fears was based on qualitative analysis of their responses to the questionnaire. The initial development of themes began with the analysis of the first responses; this thematic classification was progressively revised and completed as we read and reread (XF and MR) all the questionnaires received. For this first step of the thematic elaboration, similar answers worded similarly were brought together in the same theme; on the other hand, we took care to respect the distinction made by each woman between different themes of hopes and themes of fears. Women’s responses were reread until the point of data saturation (no more emerging themes) [9]. A last reading enabled us to classify each participant’s responses according to the classification developed in the previous stage.

The hopes and fears expected by the surgeons were categorized with the same thematic classification developed for the women.

The Comité de Protection des Personnes Nord-Ouest IV (Institutional Review Board, Lille, France) and the Agence Nationale de Sécurité du Médicament et des Produits de Santé (the French National Agency for Medicines and Health Products Safety, #B111368-30) approved the PROSPERE study, which was registered at ClinicalTrials.gov (#NCT01637441). All patients provided written informed consent.

## 3. Results

Among the 265 women included in the PROSPERE trial, 261 (98%) reported at least one hope (mean, 2.3 hopes; 72 women reported only one hope, 79 two, 70 three, 35 four, and 5 five), and 229 (86%) at least one fear (mean, 1.5 fears; 114 women reported only one fear, 77 two, 20 three, 12 four, and 6 five). Among the 39 surgeons participating in the PROSPERE trial, 16 (41%) answered the expectations questionnaire from the viewpoint of a typical patient scheduled for prolapse repair. Characteristics of the 265 women are reported in Table 1.

### 3.1. Qualitative Results for Women

Women’s hopes (Figure 1, Table 2) were categorized into 7 themes: prolapse repair, improvement of urinary function, capacity for physical activities, sexual function, well-being or comfort, end of sensations of pain or heaviness, improvement in bowel function, and no more pads (or other).

Fears (Figure 2, Table 2) were categorized into 11 themes: failure or prolapse relapse, perioperative concerns, urinary disorders, pain, sexual problems, other fears, mesh-related complications, aggravation of symptoms, physical impairment, bowel problems, and no fears.

Table 2 reports the responses of the surgeons from the PROSPERE team who completed the same questionnaire as the women, answering as they anticipated a typical patient would.

### 3.2. Quantitative Analysis

Women’s first-ranking hopes (those put in first place by the women) involved prolapse repair (51%), improvement of urinary function (14%), and well-being (14%) (see Table 3 and Figure 3).

Second-ranked hopes were for urinary improvement (31%), capacity for physical activities (19%), and sexual function (16%). The latter was also the leading third-ranked hope (25%).

Overall (Table 3), 156 (60%) women expected prolapse repair, 103 (39%) improvement in their urinary symptoms, 74 (28%) improvement in their capacity for physical activities, and 70 (27%) improvement in their sexual function. The leading hopes for women expecting improved urinary symptoms were for their incontinence to be cured (*n* = 58) and for improvement in their overactive-bladder symptoms (*n* = 36) or obstructive voiding (*n* = 15).

The main fears ranked first by women concerned a prolapse relapse (28%), concerns about the perioperative period (21%), and urinary disorders (17%) (see Table 4 and Figure 4). The leading fears ranked second were again prolapse relapse (23%) and the perioperative period (17%), as well as pain (17%).

Overall, 87 (38%) women reported a fear of prolapse relapse, 64 (28%) a fear of the perioperative period, 63 (26%) a fear of a urinary disorder, and 40 women (17%) no particular fear. The main fear among women worried about the perioperative period (*n* = 64) concerned anesthesia (*n* = 40).

The surgeons anticipated hopes and fears very similar to those the women reported (Table 3 and Table 4).

The surgeons’ responses about the preoperative hopes and fears of a typical patient were very similar to the women’s responses (Table 3 and Table 4). From the surgeons’ perspective, the first hope of a typical patient was correction of the prolapse (11 of 16 surgeons put this hope in first place) and the second was improved urinary function (6 of 16 put this hope in second place; Table 3). As the first fear of a typical patient, 8 surgeons chose the risk of failure or relapse, and as the second fear, concerns about the operation (Table 4).

## 4. Discussion

Women awaiting a cystocele repair reported as their main hopes the repair of their prolapse, improvement in their urinary symptoms, capacity for physical activities, sexual function, well-being, and end of pain or heaviness. Their main fears were related to prolapse relapse, perioperative concerns, urinary disorders, pain, and sexual problems. These reasonable expectations were consistent with the information the surgeons had given them. Our results show that surgeons were able to anticipate typical hopes and fears.

The prospective collection of these hopes and fears—before the surgery—is one of the strong points of our work, for it enabled us to measure the quality of the information as the women perceived it. The other strong point was that the information was standardized and formalized by its delivery in a leaflet and by the woman’s signature consenting to participate in the study.

One of the limitations of our work is the surgeons’ low response rate to this questionnaire. It is nonetheless consistent with the response rate expected in surveying these specialists [10]. We assume that this low response rate does not constitute a substantial bias, given that the members of the PROSPERE group habitually work together and regularly exchange information about their surgical practices [11]. Moreover, the PROSPERE investigators were involved in drafting the information leaflet and participated in conversations about every element of this information. However, it is possible that some surgeons did not respond because they did not feel sufficiently aware of each patients’ expectations to be able to respond appropriately. Another limitation is that we asked the surgeons about what they expected a typical patient would hope and fear; it would have been preferable to ask their opinion for each woman they included.

Our preliminary hypothesis was that some women might have hopes or fears that were exaggerated and did not correspond to the medical reality of these operations. On the contrary, we observed that they had reasonable expectations of the prolapse surgery: only 60% expected the complete disappearance of the prolapse, and one-third (38%) feared the risk of failure or relapse. These results are consistent with the high failure rate related to prolapse surgery [12], and suggest that surgeons do take the time to explain the risk of failure.

Alongside this focus on the anatomical results of prolapse surgery, many of the women’s expectations concerned functional results, with a quarter or more hoping for improvement in their urinary symptoms, physical capacity, sexual function, and comfort. Although the participants were aware of mesh-related complications (the study of which was the PROSPERE trial’s main objective, and which were clearly mentioned in the information sheet), less than 5% listed this specific fear.

Although studies show that POP has a negative impact on women’s sexual function [13] and that surgical management significantly improves it [14], we found that expectations of its improvement were not the women’s main priority. This prudence of the expectations of the women in our study is consistent with the results of a study showing that after surgery for POP, women’s expectations concerning physical symptoms and activity were met more often than those concerning their sexual function [15].

The French women included in our study reported expectations similar to those of the Dutch women questioned by Lawndy et al., although there were some differences in rankings [16]. While prolapse repair was the main hope (60%) in our study, Lawndy’s participants (70%) ranked the improvement of urinary symptoms first. The main fear in our study was failure or relapse (38%), while becoming incontinent was the main fear (52%) among the Dutch women. These variations may reflect differences in patient information or in the social and cultural environment between French and Dutch practices.

Surgeons who participated in the study successfully anticipated the hopes and fears reported by most women, probably because they practice this type of surgery on a regular basis. However, our study shows that the hopes and fears of women scheduled for POP surgery may vary broadly. A study analyzing interviews after surgery shows that women’s expectations focus principally on the disappearance of their physical symptoms [17]. Another work shows that women’s expectations can be multiple and concern domains as varied as urinary or gastrointestinal symptoms, physical activity, sexual function, and self-image [18].

The PROSPERE trial shows that the women who expressed preoperative expectations related to improvement of bulge symptoms were those most likely to be satisfied after the surgery [19]. We believe that our report on women’s typical expectations before prolapse repair is helpful information for physicians, but our results cannot replace an individual assessment of expectations before surgery. Anticipating typical expectations should help physicians in their individual assessment before surgery (and not replace it). As part of a patient-centered approach and individualized care pathways, identifying some expectations and fears could improve treatment adherence and patient satisfaction. Improving the doctor–patient relationship could therefore improve overall postoperative results. The quality of the preoperative information could also be integrated into this approach.

Our results support an individual patient’s hope and fear assessment before each surgical decision. Further work in this area is welcome to promote understanding and compliance with our patients’ expectations in routine clinical practice.

## 5. Conclusions

Our study may help surgeons to consider women’s expectations before pelvic-floor reconstructive surgery.

## Figures and Tables

**Figure 1 jcm-12-01332-f001:**
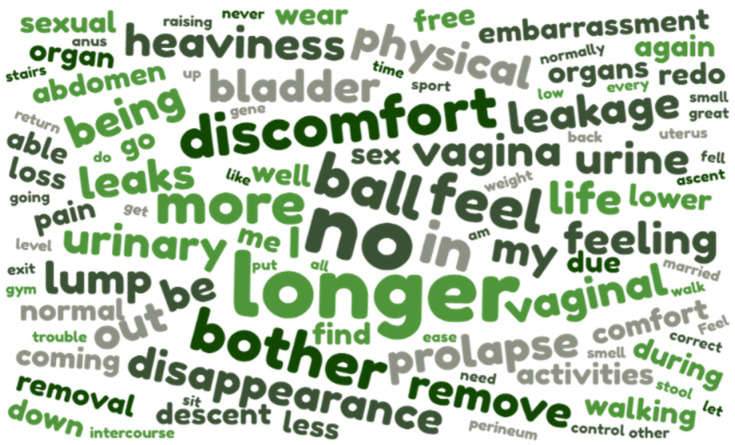
Women’s preoperative hopes before cystocele repair.

**Figure 2 jcm-12-01332-f002:**
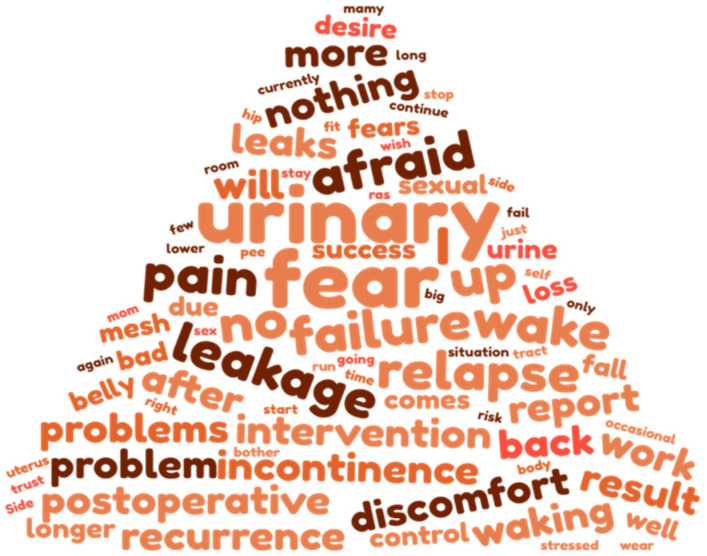
Women’s preoperative fears before cystocele repair.

**Figure 3 jcm-12-01332-f003:**
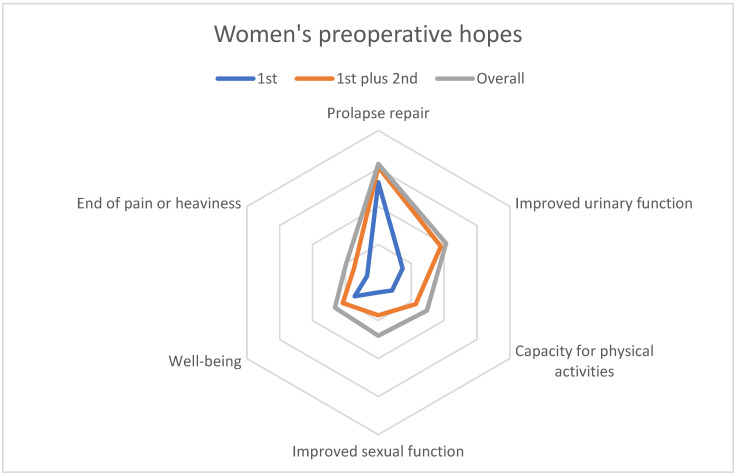
Women’s preoperative hopes.

**Figure 4 jcm-12-01332-f004:**
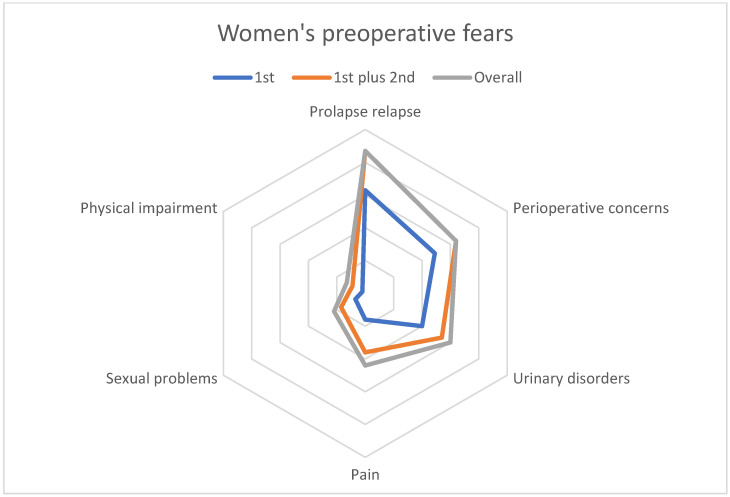
Women’s preoperative fears.

**Table 1 jcm-12-01332-t001:** Description of the population before cystocele repair.

Women Characteristics		N = 265
Age	median (range)	64 (45–75)
Number of deliveries	median (range)	2 (0–16)
POP-Q Stage—no. of patients (%)	2	58 (21.9%)
	3	185 (69.8%)
	4	21 (7.9%)
PFDI 20	means (±SD)	94.32 (±50.67)
	POPDI-6	43.81 (±22.24)
	DDI-8	16.25 (±16.18)
	UDI-6	34.25 (±25.98)
EQ5D	means (±SD)	0.78 (±0.20)

**Table 2 jcm-12-01332-t002:** Hope and Fears thematic classification for women and surgeons (examples of responses).

Hope Themes	Women	Surgeons
Prolapse repair	“no longer bulge out of my vagina”, “do not see this ball anymore”, “that my bladder is put back in place”, “stop having to push up my uterus”, “avoid this organ descent sensation”, “do not push back the ball in the vagina”	“no more vaginal ball”
Improved urinary function	“no more leakage”, “not have to go urinate every 5 min”, “fewer runs to the bathroom”, “urinate normally”	“reduction of urgency-pollakiuria”, “urinate easily”
Capacity for physical activities	“to return to normal physical activity”, “to walk and move normally”, “to regain my mobility”	“recovery of physical activity”
Improved sexual function	“resume my sex life”, “regain my femininity”	“more harmonious sexuality”
Well-being or comfort	“feel better”, “have a normal life”, “not have this embarrassment anymore”	“feel better”
End of pain or heaviness	“no more feeling of heaviness”, “no more pain in the lower abdomen”	“no longer feeling of heaviness”
Improved bowel function	“move my bowels correctly”, “no longer sitting on the toilet waiting for a bowel movement”	“move my bowels easily”
No more pads or other expectation	“not wearing pads anymore”, “no more irritation or burn”, “that the operation goes well”	-
**Fears themes**		
Failure or prolapse relapse	“fear of relapse”, “afraid that the net does not hold”	“risk of recurrence”, “failure”
Perioperative concerns	“fear of anesthesia”, “afraid of not waking up”, “afraid of nausea on waking”, “postoperative problem”, “surgical complications”	“complication during the operation”
Urinary disorder	“being incontinent”, “continuing to smell the odor of pee!”	“risk of urinary leakage”
Pain	“fear of pain”	“fear of pain”
Sexual issues	“decreased sexual desire”, “won’t recover sexual desire”	“pain during sex”
Other fear	“that my belly will be even bigger”, “infection”, “health disorder”, “gynecologic problem”, “sequelae”	“infection”
Physical impairment	“Fear of not being able to resume my physical activities”, “stay physically diminished”	“duration of convalescence”
Mesh-related complications	“mesh exposure”, “fear the implant will be rejected”	“Serious adverse effect due to the mesh”
Worsened condition	“have to pay attention to everything”, “being worse after than before”, “fear of not recovering my general fairly satisfactory condition”	“worse than before”
Bowel problems	“constipation”, “afraid of more severe constipation”	“Improvement in anorectal disorders“
No fears	“I’m not afraid, he knows what to do”, “I am confident that this operation will make me more comfortable”	-

**Table 3 jcm-12-01332-t003:** Preoperative hopes reported by women and by surgeons (for a “typical” patient).

Sample	Women (w)/Surgeons (s)						
**Hopes**	1st **N_w_**/N_s_ **261**/16 **n_w_**/n_s_	2nd **N_w_**/N_s_ **189**/16 **n_w_**/n_s_	3rd **N_w_**/N_s_ **110**/16 **n_w_**/n_s_	4th **N_w_**/N_s_ **40**/16 **n_w_**/n_s_	5th **N_w_**/N_s_ **5**/11 **n_w_**/n_s_	Overall * **Women** **N_w_ = 261** **n_w_** (**%**)	Overall * Surgeons N_s_ = 16 n_s_ (%)
Prolapse repair	**132**/11	**20**/1	**9**/1	**1**/.	**.**/.	**156** (**60**)	13 (81)
Improved urinary function	**37**/.	**58**/6	**21**/4	**6**/6	**.**/.	**103** (**39**)	15 (94)
Capacity for physical activities	**21**/.	**36**/4	**23**/3	**11**/2	**3**/3	**74** (**28**)	11 (69)
Improved sexual function	**13**/.	**30**/1	**29**/4	**4**/2	**1**/2	**70** (**27**)	9 (56)
Well-being or comfort	**36**/5	**18**/2	**16**/1	**7**/1	**2**/2	**66** (**25**)	10 (62)
End of pain or heaviness	**17**/.	**19**/2	**7**/2	**7**/1	**.**/1	**49** (**19**)	5 (31)
Improved bowel function	**3**/.	**7**/.	**4**/1	**3**/3	**.**/3	**16** (**6**)	7 (44)
No more pads or other	**1**/.	**1**/.	**2**/.	**.**/1	**.**/.	**3** (**1**)	1 (6)

* Redundant answers in the same theme were not counted; as it was possible to declare several hopes or fears, the sum of the percentages is greater than 100%. Women’s responses in bold, **N_w_**/**n_w_** for number of women responding (**N_w_**) in each theme (**n_w_**), N_s_/n_s_ for surgeons.

**Table 4 jcm-12-01332-t004:** Preoperative fears reported by women and surgeon (for a “typical” patient).

Sample	Women (w)/Surgeons (s)						
**Fears**	1st **N_w_**/N_s_ **229**/16 **n_w_**/n_s_	2nd **N_w_**/N_s_ **115**/16 **n_w_**/n_s_	3rd **N_w_**/N_s_ **38**/16 **n_w_**/n_s_	4th **N_w_**/N_s_ **18**/15 **n_w_**/n_s_	5th **N_w_**/N_s_ **6**/11 **n_w_**/n_s_	Summary * **Women** **N_w_ = 229** **n_w_** (**%**)	Summary * Surgeons N_s_ = 16 n_s_ (%)
Failure or prolapse relapse	**63**/8	**27**/2	**4**/1	**.**/2	**2**/3	**87** (**38**)	15 (94)
Perioperative concerns	**49**/5	**20**/5	**7**/4	**3**/.	**1**/.	**64** (**28**)	14 (87)
Urinary disorders	**40**/1	**14**/4	**7**/8	**.**/2	**.**/.	**60** (**26**)	14 (87)
Pain	**16**/1	**20**/3	**5**/2	**4**/4	**.**/4	**44** (**19**)	10 (62)
Sexual problems	**7**/1	**10**/.	**2**/.	**2**/3	**1**/3	**22** (**10**)	7 (44)
Other fear	**2**/.	**7**/.	**7**/1	**6**/1	**2**/2	**17** (**7**)	3 (19)
Physical impairment	**2**/.	**7**/.	**4**/.	**.**/1	**.**/1	**13** (**6**)	1 (6)
Mesh-related	**5**/.	**4**/1	**1**/.	**1**/1	**.**/.	**11** (**5**)	2 (12)
Worsened condition	**4**/.	**2**/1	**1**/.	**1**/.	**.**/.	**7** (**3**)	2 (12)
Bowel problems	**1**/.	**3**/.	**.**/.	**1**/1	**.**/1	**5** (**2**)	1 (6)
No fears	**40**/.	**1**/.	**.**/.	**.**/.	**.**/.	**40** (**17**)	.

* Redundant answers in the same domain were not counted; as it was possible to declare several hopes or fears, the sum of the percentages is greater than 100%. Women’s responses in bold, **N_w_**/**n_w_** for number of women responding (**N_w_**) in each theme (**n_w_**), N_s_/n_s_ for surgeons.

## Data Availability

Data from the analysis are available upon request from the authors.

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
