# Peer review of "Do Surgeons Anticipate Women’s Hopes and Fears Associated with Prolapse Repair? A Qualitative Analysis in the PROSPERE Trial"

_jcm, 2023, doi:10.3390/jcm12041332_

Round 1

Reviewer 1 Report

This interesting study aimed to qualitatively assess hopes and fears of women submitted to POP surgery after preoperative medical interview in the contest of the PROSPERE trial, and to compare those with hopes and fears of POP surgeons.

The study design is simple and difficult to improve due to the qualitative assessment of the research; a standardization of the questions asked to the patients lacks and this condition does not help to generalize reported data.

However, this topic can be really interesting for surgeons performing POP surgery and can really help physicians in the preoperative assessment of the patients. I think that one of the main aims of the authors was to highlight how a complete and exhaustive preoperative interview could be really helpful in establishing a good connection between patient and physicians, and share goals of the surgery.

I have just some minor revisions for the authors: 

Mathods

  • 45-75 years-old is a really wide range of age. Do you think that this could affect the number and the quality of answers? Do you think that a stratification by age could be possible or it was too difficult to perform in relation to the sample size?

  • A qualitative analysis is difficult to evaluate because of a lack of standardize and validate questions. Moreover, authors ranked the hopes and fears of the patients. Even if there was a double check, this should be considered as a bias of the study. 

  • Line 94: I would add symptomatic 2 stage prolapse in inclusion criteria.

  • Have surgeons expressed their hopes and fears before the categorization of patients’ answers or after? If they answered after it could be a bias and it should be signaled.

Results

  • Re-edit table 1 please. 

  • Figure 1 and 2 I think they’re not necessary.

  • Line 168: Do you mean table 3?

  • Re-edit table 3 (write in bold the heading).

  • Line 175: Do you mean table 3?

  • Line 184: Do you mean table 4?

  • Line 197: According to the table the main hope for surgeons is about urinary symptoms and not POP. Prolapse repair is not 12/16 but 13/16 according to table 3.

Discussion

  • Sexual function is a wide topic and generally multifactorial. Do you think that an age correlation would help the analysis of data?

Author Response

Response to reviewer 1

1) 45-75 years-old is a really wide range of age. Do you think that this could affect the number and the quality of answers? Do you think that a stratification by age could be possible or it was too difficult to perform in relation to the sample size?

Thanks for this suggestion, we keep it for further analysis. Our initial aim was descriptive.

2) A qualitative analysis is difficult to evaluate because of a lack of standardize and validate questions. Moreover, authors ranked the hopes and fears of the patients. Even if there was a double check, this should be considered as a bias of the study.

It is a misunderstanding. Women rank themselves their hopes and fears (not us). We analysed women responses to identify the themes. This is in line with the usual methodology of a thematic analysis [Srikrishna S, Robinson D, Cardozo L, Cartwright R. Experiences and expectations of women with urogenital prolapse: a quantitative and qualitative exploration. BJOG 2008;115:1362–8].

We completed the methods chapter (lines 121-2) to clarify these points: “Women’s responses were reread until the point of data saturation (no more emerging themes) [9]”

3) Line 94: I would add symptomatic 2 stage prolapse in inclusion criteria.

Thank you, precision added line 94.

4) Have surgeons expressed their hopes and fears before the categorization of patients’ answers or after? If they answered after it could be a bias and it should be signaled.

Yes, they were blinded of women’s responses. We added this point line 111.

5) Re-edit table 1 please.

Thanks, done.

6) Figure 1 and 2 I think they’re not necessary.

This descriptive mode of presentation is the one usually used in qualitative studies. We believe that to promote citation of the manuscript, graphical representations of the results are relevant. We leave the choice to the editor.

7) Line 168: Do you mean table 3?

Thanks, correction done.

8) Re-edit table 3 (write in bold the heading).

Done.

9) Line 175: Do you mean table 3?

Thanks, correction done.

10) Line 184: Do you mean table 4?

Thanks again, correction done.

11) Line 197: According to the table the main hope for surgeons is about urinary symptoms and not POP. Prolapse repair is not 12/16 but 13/16 according to table 3.

It is a misunderstanding. Table 3 present all surgeons responses. Our sentence was about the first hope anticipated by the surgeons. Precision added lines 200-1.

12) Sexual function is a wide topic and generally multifactorial. Do you think that an age correlation would help the analysis of data?

The presentation of results in the form of summary tables and averages should not obscure the wide variety of individual expectations (regardless of age). We believe that an individual patient’s hope and fear assessment is required before each surgical decision (see line 269).

Reviewer 2 Report

A very interesting study which is relevant the practicing surgeon. The methodology was well thought out in genera, except for the limitation presented in the discussion (the limited number of surgeons who completed  their survey). However, I would agree with the authors that the more significant issue was a recognition of what motivates or inhibits patients decisions for surgical management.  I do have suggestions to improve the manuscript:  a.  specific statement on how future work in this area can be used to address patient concerns b. were the expectations and concerns different in the two groups of the i.e between the mesh and native tissue groups. c: it would be very interesting to know if patients fears and expectations were related to their subjective feelings about the outcome of the procedure. This may be addressed in the main study, but some reference to this important topic would be useful in this study as well. 

Author Response

Response to reviewer 2

A very interesting study which is relevant the practicing surgeon. The methodology was well thought out in genera, except for the limitation presented in the discussion (the limited number of surgeons who completed  their survey). However, I would agree with the authors that the more significant issue was a recognition of what motivates or inhibits patients decisions for surgical management.  I do have suggestions to improve the manuscript:  

  1.  specific statement on how future work in this area can be used to address patient concerns.

We added a sentence at the end of the discussion, lines 269-71: “Our results support an individual patient's hope and fear assessment before each surgical decision. Further work in this area is welcome to promote understanding and compliance with our patients' expectations in routine clinical practice”.

  1. were the expectations and concerns different in the two groups of the i.e between the mesh and native tissue groups.

All patient included underwent a surgery with mesh by laparoscopy or by the vaginal route, modification done lines 88-9.

c: it would be very interesting to know if patients fears and expectations were related to their subjective feelings about the outcome of the procedure. This may be addressed in the main study, but some reference to this important topic would be useful in this study as well. 

We showed that women included in the PROSPERE trial who expressed preoperative expectations related to improvement of bulge symptoms were those most likely to be satisfied after the surgery (Chattot C, Deffieux X, Lucot JP, Fritel X, Fauconnier A. Preoperative predictors and a prediction score for perception of improvement after mesh prolapse surgery. Int Urogynecol J 2020;31:1393–400). See discussion lines 260-2.

Reviewer 3 Report

The authors give explicit data on patients’ expectations and fears regarding prolapse surgery, which could be of clinical interest for pelvic surgeons. However, the manuscript has some aspects that need to be reviewed before consider its publication.

Results

  1. The study shows a correlation between patient’s hope and fears and surgeons’ believing on these expectations from a “typical patient”. Therefore, the results show that patient’s expectations correlate to presumed women expectations by surgeons in a “typical case”. However, the statement “we found and excellent correlation between women’s and surgeon’s ranking of both hopes and fears” (lines 207-209) is not appropriate.
  2. Please, consider if this correlation analysis is indicated, as, in my opinion, it gives no relevant data.

Discussion

  1. Limitations, lines 226-234. Please, give some explanation to low response rate from surgeons to questionnaires. Why surgeons that are involved in the leaflet design (as stated by the authors) did not response to questionnaires of their own study? Have the authors contacted the collaborating surgeons regarding this question?
  2. The authors assumption that surgeon’s low response rate would not suppose a substantial bias is not acceptable when the majority of surgeons (59%) have not answered the questionnaires. Please, state the potential bias on this regard.
  3. The limitation on surgeons asked for expected answers on “typical patients” needs to be emphasized. This is of special interest in the case of functional surgery centred on improve woman’s QoL that needs an individualized approach.

Figures

  1. Figures 1 and 2 seem indicated for an oral presentation. However they do not give any relevant data for the manuscript and enlarge the article unnecessarily. Please, consider remove them.

Author Response

Please find attached our response to reviewer 3 comments

and the latest version of our manuscript

Many thanks

Round 2

Reviewer 3 Report

The authors have answered satisfactorily to all questions raised.